# Cancer Stem Cells in Oral Squamous Cell Carcinoma: A Narrative Review on Experimental Characteristics and Methodological Challenges

**DOI:** 10.3390/biomedicines12092111

**Published:** 2024-09-16

**Authors:** Surendra Kumar Acharya, Saptarsi Shai, Yee Fan Choon, Indrayadi Gunardi, Firstine Kelsi Hartanto, Kathreena Kadir, Ajoy Roychoudhury, Rahmi Amtha, Vui King Vincent-Chong

**Affiliations:** 1Department of Oral Medicine, Radiology and Surgery, Faculty of Dentistry, Lincoln University College, Petaling Jaya 47301, Selangor, Malaysia; 2Center for Cell and Gene Therapy, Baylor College of Medicine, Texas Children’s Hospital, Houston, TX 77030, USA; saptarsi.shai@bcm.edu; 3Department of Oral and Maxillofacial Surgical Sciences, Faculty of Dentistry, MAHSA University, Jenjarom 42610, Selangor, Malaysia; choonyeefan@yahoo.com; 4Oral Medicine Department, Faculty of Dentistry, Universitas Trisakti, Jakarta 11440, Indonesia; indrayadi@trisakti.ac.id (I.G.); firstine@trisakti.ac.id (F.K.H.); 5Department of Oral and Maxillofacial Clinical Sciences, Faculty of Dentistry, University of Malaya, Kuala Lumpur 50603, Malaysia; kathreena@um.edu.my; 6Department of Oral and Maxillofacial Surgery, All India Institute of Medical Sciences, New Delhi 110029, India; ajoyroy@hotmail.com; 7Department of Oral Oncology, Roswell Park Comprehensive Cancer Center, Buffalo, NY 14263, USA

**Keywords:** cancer stem cell (CSC), oral squamous cell carcinoma (OSCC), CSC signaling, CSC therapy

## Abstract

Cancer stem cells (CSCs) represent a subpopulation of cancer cells that are believed to initiate and drive cancer progression. In animal models, xenotransplanted CSCs have demonstrated the ability to produce tumors. Since their initial isolation in blood cancers, CSCs have been identified in various solid human cancers, including oral squamous cell carcinoma (OSCC). In addition to their tumorigenic properties, dysregulated stem-cell-related signaling pathways—Wnt family member (Wnt), neurogenic locus notch homolog protein (Notch), and hedgehog—have been shown to endow CSCs with characteristics like self-renewal, phenotypic plasticity, and chemoresistance, contributing to recurrence and treatment failure. Consequently, CSCs have become targets for new therapeutic agents, with some currently in different phases of clinical trials. Notably, small molecule inhibitors of the hedgehog signaling pathway, such as vismodegib and glasdegib, have been approved for the treatment of basal cell carcinoma and acute myeloid leukemia, respectively. Other strategies for eradicating CSCs include natural compounds, nano-drug delivery systems, targeting mitochondria and the CSC microenvironment, autophagy, hyperthermia, and immunotherapy. Despite the extensive documentation of CSCs in OSCC since its first demonstration in head and neck (HN) SCC in 2007, none of these novel pharmacological approaches have yet entered clinical trials for OSCC patients. This narrative review summarizes the in vivo and in vitro evidence of CSCs and CSC-related signaling pathways in OSCC, highlighting their role in promoting chemoresistance and immunotherapy resistance. Additionally, it addresses methodological challenges and discusses future research directions to improve experimental systems and advance CSC studies.

## 1. Introduction

In 2023, head and neck (HN) cancer, including the oral cavity and pharynx, remained the tenth most common malignancy worldwide among men, with more than 90% of these malignancies diagnosed as squamous cell carcinoma (SCC), known as HNSCC [1]. In South and Southeast Asia and the Western Pacific, cancer of the lip and oral cavity was a leading cause of cancer-related deaths among men. This type of cancer ranked 16th globally in both incidence and mortality, underscoring its significant impact on public health in these regions [2]. Standard conventional treatments for oral squamous cell carcinoma (OSCC) patients include surgery, radiotherapy, and chemotherapy [3,4]. Despite the use of adjuvant therapies, such as platinum-based chemotherapy/5-flurorouracil (5-FU) and cetuximab, locoregional recurrences and metastasis remain major factors for a dismal 5-year survival rate of 50% [5,6,7,8]. In addition, the economic and cost burden of treating head and neck cancer is significantly higher than other cancers [9]. The recent identification of a highly tumorigenic subgroup of cancer cells has received enormous attention [10,11,12,13,14,15]. These cells are known as cancer stem cells (CSCs), and they are postulated to be responsible for the recurrences, metastasis, and poor prognosis of cancers, including OSCC [16,17,18]. Elimination of CSCs theoretically improves the prognosis of the disease [19,20]. Indeed, many small molecules inhibitors targeting CSC-signaling pathways have been developed and are in different phases of clinical trials. Some of them, for instance, vismodegib and glasdegib, targeting smoothened (Smo) protein of the hedgehog signaling pathway, have been approved by the United States (U.S.) Food and Drug Administration (FDA) and launched for the treatment of basal cell carcinoma and acute myeloid leukemia, respectively [21,22,23]. Such advancement in treatment, however, is not observed in patients with OSCC, even though ample studies have demonstrated the existence of CSCs in OSCC. One of the proposed contributing factors for the slow progress in novel treatment in OSCC could be due to the lack of specificity in markers that have been utilized to isolate and characterize CSCs in OSCC [24]. The aim of this narrative review is, therefore, to present experimental evidence on the existence of CSCs in OSCC and discuss potential experimental systems to overcome methodological challenges to better understand CSCs in OSCC.

### Search Strategy

Databases, including PubMed, Google Scholar, Ebsco, and Science Direct, were searched from 2007 (first report of CSCs in HNSCC) to 2024 using various combinations of the following keywords: “cancer stem cells in oral squamous cell carcinoma”, “cancer stem cells in head and neck squamous cell carcinoma”, “oral cancer stem cells”, “cancer stem-like cells”, “oral squamous cell carcinoma, OSCC”, “head and neck squamous cell carcinoma, HNSCC”, “cancer stem cells”, “cancer stem cells markers,”, “cancer stem cells pathways,” “chemoresistance and radioresistance in OSCC”, “chemoresistance and radioresistance in HNSCC”, “cancer stem cells targeted therapies”, and “cancer stem cells therapies”. Original experimental studies (both in vitro and in vivo), reviews, editorial letters, book chapters, opinions, and abstracts from the analyses published in English considering stem cell markers in HN-/OSCC were considered and included.

## 2. Cancer Stem Cells in Oral Squamous Cell Carcinoma

### 2.1. The CSCs Model

A couple of cancer evolution models have been put forth to elucidate heterogeneity in the cancer cell population: the CSCs model and Nowell’s clonal evolution model. As depicted in Figure 1, in the CSCs model, the cancer cell population is proposed to comprise (1) CSCs and (2) non-CSCs. The CSCs behave like normal stem cells in that they can either divide symmetrically through a self-renewal mechanism to propagate more CSCs or asymmetrically to give rise to more differentiated cells, i.e., non-CSCs. Hence, a malignant cellular hierarchy exists in the CSCs model in which the CSCs that occupy the apex are tumorigenic, as they continuously propagate tumors, in addition to giving rise to a differentiated, non-tumorigenic population, non-CSCs [16,25,26,27,28,29,30,31]. However, no such hierarchy exists in Nowell’s clonal evolution model. In this model, heterogeneity of the cancer cell population is attributed to the clonal expansion of cancer cells after acquiring genetic mutations, resulting in different phenotypic characteristics [32,33].

### 2.2. Definition of CSCs

As such, the CSCs model theoretically implies that, as long-lived resident cells in the normal tissues, stem cells are the targets for accumulation of genetic aberrations responsible for malignant transformation (Figure 1) [4,25,34]. Indeed, when induced with a carcinogen, 4-nitroquinoline 1-oxide (4-NQO), cell lineage tracing showed that normal stem cells in the basal layer of the tongue epithelium of transgenic mice were the cells of origin of carcinogenesis [35]. However, according to the American Association for Cancer Research (AACR) Workshop on CSCs, CSCs are not necessarily derived from normal stem cells [29], as tumors have been shown to arise from some “progenitor” or differentiated cells [36,37,38,39]. The AACR Workshop on CSCs defines CSCs as cancer cells that produce tumors; hence, xenotransplantation assay is the gold standard to demonstrate the tumorigenicity of CSCs. Cancer cells that are tumorigenic and form tumors when injected into mouse models are CSCs. Therefore, CSCs are defined only by xenotransplantation assay regardless of the cell of origin [29].

### 2.3. In Vivo Evidence for CSCs

Thus defined, CSCs were first demonstrated in human acute myeloid leukemia (AML). By using cell surface markers, CD34 and CD38, which identified progenitor and pluripotent stem cells in bone marrow, AML-initiating cells that produced leukemia when transplanted in immunodeficient mice were observed to be CD34^+^CD38^−^ cells. Furthermore, CD34^+^CD38^+^ cells failed to produce leukemia in immunodeficient mice. Hence, CD34^+^CD38^−^ cells fulfilled the definition of CSCs. Additionally, CD34^+^CD38^−^ cells were able to differentiate into more differentiated leukemic blasts, indicating the retention of the cellular hierarchy in AML proposed by the CSCs theory [11,12]. After this successful demonstration of CSCs in AML, marker-based identification of CSCs is routinely used in CSCs study for solid tumors, as depicted in Figure 2. CSCs from solid tumors are identified by using fluorescence- or magnetic-beads-conjugated putative CSCs markers, such as CD44 or CD133. CSCs are then sorted and examined for in vitro characteristics, such as holoclone and sphere formations and, most importantly, tumor formation in xenotransplantation assay, as it is the gold standard to demonstrate CSCs [29]. With this approach, CSCs were subsequently demonstrated in breast cancer [13], glioblastoma [40], colorectal cancer [41], pancreatic cancer [42], and ovarian cancer [43]. The same methodological approach was used to demonstrate the existence of CSCs in HN- and OSCC. Previously reported putative CSCs markers for other solid tumors (CD44 in breast cancer and CD133 in brain tumor) were used to identify HN-/OSCC CSCs and then isolated and xenotransplanted in mice to observe for tumor formation. Table 1 chronologically summarizes the in vivo evidence of CSCs in HN- and OSCC from the first report in 2007 until 2024 [44,45,46,47,48,49,50,51,52,53,54,55,56,57,58,59,60,61,62,63,64,65,66,67,68].

### 2.4. In Vitro Characteristics of CSCs

#### 2.4.1. Morphology

A monolayer culture of normal keratinocytes generated three different colony morphologies: (1) the holoclones, which are compact round colonies; (2) the paraclones, which are loose irregular colonies; and (3) the meroclones, which exhibit intermediate features; and these clones are derived from stem-, early-, and late-amplifying keratinocytes, respectively [69,70]. In the culture of human and mouse tongue epithelial cells, the results showed that holoclones were the stem cell compartment, as they readily generated holoclones at each passage, confirming a self-renewal property [71,72]. Such clonogenic morphologies were also reported in HN-/OSCC cell lines. Holoclones of HN-/OSCC cell lines also showed a stem cell property, as they were able to generate holoclones upon repassing. Furthermore, when examined for expression of putative HN-/OSCC stem cell markers like CD44 and CD133 (discussed in Section 2.4.3), the holoclones overexpressed these markers, but this expression was weak in the meroclones and paraclones. Using the marker-based sorting method as described in Figure 2, fresh HNSCC tumor samples derived from CD44^+^ cancer cells also formed holoclones, while CD44^−/low^ cancer cells formed paraclones, indicating the retention of a malignant cellular hierarchy in HNSCC [50,73]. Holoclone formation is a consistent morphological feature of CSCs [74,75,76,77,78,79].

#### 2.4.2. Sphere Formation

In a neural stem cells study, it has been shown that neural stem cells, such as ependymal cells, were able to proliferate as free-floating spheres, which differentiated into neurons, astrocytes, and oligodendrocytes when transferred to an adhesive substrate [80,81,82]. Hence, the ability to proliferate as spheres in non-adherent culture is used to assess stem cell activity [83,84]. Chen and colleagues showed that spheres generated from OSCC cell lines in a non-adhesive culture system were CSCs, as these spheres were shown to express stem cell markers, CD133 and aldehyde dehydrogenase (ALDH1) (discussed in Section 2.4.3), exhibit chemo- and radioresistance, and, more importantly, produce tumors in mice [85]. Moreover, marker-identified CSCs in HNSCC and OSCC have also been reported to consistently form spheres in suspension culture [86,87]. As such, sphere-forming assays are increasingly being used as an important tool to assess the potential of cells with stem cell traits, and, as will be discussed in details in Section 5, three-dimensional (3D) cell culture methods such as this may provide an alternative method for detecting CSCs than the conventional marker-based approach [88].

#### 2.4.3. Markers

The use of markers has been the cornerstone in CSCs research. They are the means of identification, isolation, enrichment, and characterization of CSCs [89,90]. Evidently, a plethora of markers have been used to isolate CSCs from either primary OSCC tumors or cell lines (refer to Table 1). Here, we focus on and delineate the top three markers highlighted in a systemic review [91]: (a) CD44, (b) ALDH, and (c) CD133, which were routinely used to isolate and characterize CSCs in OSCC.

(a)CD44

CD44 is a non-kinase cell surface glycoprotein that has been used as a marker for CSCs [92]. Based on CD44 expression, CSCs in breast cancer were demonstrated. As few as 100 CD44^+^CD24^−/low^Lineage^−^ cells formed tumors in mice as compared to the alternate phenotype [13]. Replicating this method of identification in breast cancer, CSCs in HNSCC were first reported by Prince and colleagues in 2007. The group showed that a distinct population of CD44^+^ and CD44^−^ cancer cells was identifiable from resected primary HNSCC tumors. They demonstrated that CD44^+^ cells were CSCs, as they formed tumors in mice. In addition, CD44^+^ cancer cells also differentially expressed a stem-cell-related gene implicated in tumorigenesis, polycomb complex protein BMI-1 (*BMI1*) [44]. Since then, CD44 has become one of the most common putative markers used to identify CSCs in HN- and OSCC, either singly or in combination with other markers (discussed below) [93] (refer to Table 1). In a study to ascertain the role of CD44 expression in the survival of OSCC patients, Oliveira and colleagues reported that the absence of CD44 (and CD24) was associated with a better overall survival rate [94]. In contrast, expression of CD44 (and ALDH1, a phosphorylated signal transducer and activator of transcription 3 known as p-STAT3) was reported to be associated with a poor prognosis for HNSCC patients [48]. Alternative splicing of CD44 gene generates two families of CD44 isoforms: CD44v and CD44s [92]. To examine if CD44 merely serves as a marker of CSCs or plays a role in conferring CSCs traits to cancer cells, Zhang and colleagues showed that CD44s is strongly expressed in CSCs, and by knocking down CD44, the tumorigenicity of cancer cells was greatly reduced, with a concomitant downregulation of CD44 expression in these cells. Reintroduction of CD44s into cancer cells restored their tumor growth capacity, suggesting that CD44 is not merely a marker but also plays an essential role in promoting CSC traits [95]. Recently, Bai et al. [96] demonstrated that knocking down ephrin receptor A2 (EphA2) attenuated the CSC phenotype in OSCC by inhibiting the expression of CD44 and CD133 via the EphA2/Krüppel-like factor 4 (Klf4) axis in vitro. Additionally, one of the CD44 variants, CD44v3 (and CD24low) phenotype, was significantly correlated with post-operative lymph node metastasis and local recurrence in OSCC [97].

(b)Aldehyde dehydrogenase (ALDH)

ALDH is an intracellular enzymatic family consisting of 19 isoforms that are localized in the cytoplasm, mitochondria, or nucleus and are responsible for oxidizing aldehydes to carboxylic acids [98]. Several of its isoforms were involved in retinoic acid (RA) cell signaling via RA production by oxidation of all-trans-retinal and 9-cis-retinal, which has been linked to the “stemness” of CSCs [99,100,101,102]. Of the 19 isoforms, expression of ALDH1 was reported to be a putative marker in HNSCC. ALDH1^+^-only cells possessed unequivocal CSCs traits, such as tumorigenicity, sphere formation, chemoresistance, and invasion, as CD44^+^CD24^−^ALDH1^+^ cells. High ALDH1 expression was also associated with a poor prognosis [46]. Furthermore, Clay and colleagues demonstrated that ALDH^+^ cancer cells isolated from a primary OSCC tumor were ten times more tumorigenic than CD44^+^-only cells, strongly supporting the use of ALDH as a more discriminatory marker for CSCs in HN-/OSCC [47]. Additionally, knocking down of ALDH3A1 significantly reduced the cancer cells’ viability in cisplatin treatment, indicating ALDH conferred chemoresistance in OSCC [103]. The method of using ALDH to detect CSCs is different from the marker-based approach described in Figure 2. The ALDEFLUOR system is used to detect ALDH activity in CSCs, in which BODIPY-aminoacetaldehyde (BAAA) is used as the ALDH substrate and N,N-diethlyamionobenzalhyde (DEAB) as the negative to detect and isolate CSCs exhibiting high ALDH activity. The ALDEFLUOR system was reportedly able to detect the activities of isoforms ALDH1A1, ALDH1A2, ALDH1A3, ALDH1B1, ALDH2, ALDH3A1, ALDH3A2, ALDH3B1, and ALDH5A1. Whether or not ALDH plays an essential role in conferring CSC traits to cancer cells like CD44 is currently unreported, but ALDH as a therapeutic target in several cancers has entered different phases of clinical trials [104]. Additionally, overexpression of ALDH1 (and p75 neurotrophin receptor, p75NTR) at the tumor invasive front was reported to be an independent predictor of decreased survival and metastasis in OSCC [105].

(c)CD133

Human CD133 (prominin-1) is a glycosylated protein with five transmembrane domains and two large extracellular loops [106]. Initially characterized as a marker for hematopoietic stem cells [107,108], CD133^+^ cancer cells isolated from human brain [14,40] and laryngeal [109] tumors were demonstrated to be CSCs. In OSCC, Chiou and colleagues demonstrated CD133 to be a CSC marker. Instead of employing CD133 in the conventional marker-based approach, the group used spheres formed (discussed in Section 2.4.2) in serum-free cultivation to assess the stem cell activity in OSCC cancer cells. Through this approach, the spheres (or oral cancer stem-like cells, OC-SLC, as they were termed) were demonstrated to be enriched with CSCs, as they more readily formed tumors in mice than the parental cells they were derived from. They also showed that these OC-SLC were stained positively for stem cell markers, such as octamer-binding transcription factor 4 (Oct4), nanog homeobox (Nanog), and CD133. Additionally, overexpression of these markers was reported to be associated with poor overall survival in oral cancer patients [45]. In another study, magnetically sorted CD133^+^ cells showed all CSC traits, such as tumor and sphere formation. These cells were also more resistant to paclitaxel, establishing CD133 as a CSC marker for OSCC [49]. Thereafter, more studies have used CD133 as a CSC marker in OSCC either alone or in combination with other markers (refer to Table 1). Moreover, CD133 may play a role in conferring CSC traits to cancer cells, as, in a study examining the function of CD133, silencing of CD133 attenuated a CSC population and tumorigenicity in OSCC. Treating CD133-knockdown CSC population with cisplatin reduced its proliferation and invasion, suggesting potential CD133-based therapies for OSCC [54].

## 3. Signaling Mechanisms of CSC

The CSC model views a tumor as an abnormal organ consisting of a hierarchy of cells, including self-renewing stem cells and highly proliferative progenitor cells, which differentiate to form the bulk of the tumor mass [16]. In HNSCC, the primary molecular characteristics of unchecked cell replication are the inactivation of the tumor suppressors p53 and retinoblastoma (RB). Additionally, frequent mutations in the epidermal growth factor receptor mitogen-activated protein kinase kinase 1 (EGFR-MEK), Notch, and phosphatidylinositol-4,5-bisphosphate 3-kinase catalytic subunit alpha/Akt serine/threonine kinase 2/phosphatase and tensin homolog (PI3K/Akt/PTEN) signaling pathways contribute to abnormal mitogenic signaling. Locoregional tumors are often amenable to surgical removal. However, many tumors are diagnosed at a locally advanced stage, which results in a poor prognosis despite multimodal treatments like surgery, radiotherapy, and chemotherapy, especially when combined with a human papilloma virus deoxyribonucleic acid (HPV-DNA)-negative status. Over 50% of patients with locally advanced HNSCC develop metastases or experience relapse, leading to survival rates of less than a year [110,111]. Preclinical and clinical findings indicate that CSCs are able to survive chemo- and radiotherapy and dynamically adapt to changing environmental conditions, e.g., hypoxia or a lack of nutrients [112,113,114]. They achieve this through various molecular pathways that support their survival and proliferation, contributing to therapy resistance. In the next section, we will delve deeper into these molecular pathways and mechanisms that enable CSCs to withstand conventional treatments and drive cancer progression. The precise regulation of stem cell functions is essential for normal biological activity. Several key developmental and signaling pathways are crucial in this regulatory process. These include the Janus-activated kinase/signal transducer and the activator of transcription (JAK/STAT), Wnt family member/beta-catenin (Wnt/β-catenin), PI3K/PTEN pathways. These pathways mediate various stem cell properties, such as self-renewal, cell fate decisions, survival, proliferation, and differentiation. It is not surprising that many of these critical pathways are dysregulated in cancer.

### 3.1. JAK/STAT Pathway

In extension to these phenomena, the JAK/STAT pathway has been shown to play a crucial role in regulating CSC. Inhibiting JAK/STAT signaling reduces the population of CSC-like cells resistant to the chemotherapy drug doxorubicin, thereby enhancing drug efficacy. However, some CSC-like cells remain resistant to doxorubicin despite JAK–STAT inhibition, suggesting the involvement of additional pathways. Microarray data highlight the importance of JAK/STAT signaling, alongside other pathways, like tumor necrosis factor alpha (TNFα) and KRAS proto-oncogene, GTPase (KRAS) signaling, in doxorubicin resistance [115]. Activation of JAK/STAT signaling in radioresistant colorectal cancer (CRC) tissues, particularly JAK2 overexpression in CRC stem cells, correlates with metastasis. JAK2/STAT3 signaling promotes tumor initiation and radioresistance by inhibiting apoptosis and enhancing clonogenic potential. Mechanistically, STAT3 binds to the cyclin D2 (CCND2) promoter, increasing its transcription and sustaining cancer stem cell growth post-radiotherapy by maintaining cell cycle integrity and reducing DNA damage accumulation. These findings suggest that JAK2/STAT3/CCND2 signaling tends to be a resistance mechanism in CRC, offering potential biomarkers and therapeutic avenues for improving outcomes in CRC patients post-radiotherapy [116].

### 3.2. Wnt/β-Catenin Pathway

The Wnt pathway plays a pivotal role in cancer stemness, with beta-catenin (β-catenin) activating telomerase reverse transcriptase (TERT) expression, promoting telomere maintenance. The leucine-rich repeat-containing G-protein coupled receptor 5 (Lrg5) marks intestinal stem cells, driving tumor growth when adenomatous polyposis coli (APC) is lost. The Rac family small GTPase 1 (RAC1) activation post-APC loss expands the Lgr5 population via reactive oxygen species/nuclear factor kappa-light-chain-enhancer of activated B cells/Wnt (ROS/NF-κB/Wnt) signaling. Tumor microenvironment (TME) factors, like hepatocyte growth factor (HGF) and periostin, enhance Wnt activity, sustaining tahe cancer stem cell phenotype. Non-coding RNAs like miR-146a stabilize β-catenin, while long non-coding transcription factor 7 (lncTCF7) activates TCF7, promoting self-renewal in liver CSCs. Intestinal stemness signatures correlate with a poor prognosis in colorectal cancer, reflecting a tumor differentiation status rather than Wnt-driven stem cell numbers [117]. Additionally, by inhibiting β-catenin signaling, XAV939 effectively curbed CSC progression in HNSCC and CRC cells, thereby mitigating CSC-induced chemical resistance [118].

### 3.3. PI3K/Akt/mTOR Pathway

The PI3K/Akt/mammalian target of rapamycin (mTOR) pathway plays a crucial role in maintaining stemness by inhibiting the MEK/extracellular signal-regulated kinases (ERK) signaling pathway. This pathway’s involvement extends beyond specific cancer types, also influencing the stemness of mesenchymal stem cells (MSCs). Stem cell markers like CD44, CD133, CD24, epithelial cell adhesion molecule (EpCAM), Lrg5, and ALDH1 are regulated by S6 kinase 1 (S6K1), a downstream component of PI3K/Akt/mTOR. Additionally, the inactivation of pathway inhibitors, such as PTEN, amplifies PI3K/Akt/mTOR signaling, leading to an expansion of the stem cell population, as observed in adult acute leukemia (ALL) [119]. In CSCs, ALDH activity, a primary property, is regulated by PI3K signaling and the sex-determining region Y-box 2 (Sox2). Sox2 expression, induced by PI3K signaling, significantly increases ALDH1A1 and the ALDH^+^ cell population by directly interacting with the ALDH1A1 promoter. Additionally, squamous tumor cells exhibit dynamic reactivation of PI3K/mTOR via EGFR/AXL receptor tyrosine kinase (AXL)/protein kinase C (PKC) following pharmacological inhibition of PI3Kα. This reactivation enhances Sox2 translation, subsequently upregulating genes associated with CSC stemness characteristics [120,121]. Moreover, Sox2 overexpression was found to restore transcriptional and translational levels of ALDH1A1, ALDH3A1, and CD44, which were suppressed after knocking down of ubiquitin-specific protease 14 (USP14), a well-established carcinogenic gene in HNSCC. The results suggested a master regulator role of Sox2 in the CSCs’ state [122]. Additionally, a study has shown PIK3CA overexpression enhances CSCs’ population in both murine and human HNSCC. However, CSC maintenance in PIK3CA-overexpressing HNSCC becomes independent of the PI3K pathway. Compensatory mechanisms, including ephrin receptors (Ephs), tropomyosin receptor kinases (TRKs), and proto-oncogene c-KIT (c-Kit) signaling activation, sustain CSCs’ population. Co-targeting these pathways may effectively eliminate PI3K-independent CSCs, offering potential for novel anti-CSC therapeutic strategies in HNSCC, particularly for patients with PIK3CA amplification [123]. Increasing evidence suggests that hyperactive or abnormal signaling within these pathways contributes to the survival of CSCs. CSCs are a relatively rare population of cancer cells with the ability to self-renew, differentiate, and generate serially transplantable heterogeneous tumors across various cancer types [124]. CSCs are located at the invasive fronts of HNSCC, particularly near blood vessels (perivascular niche). Signaling events initiated by endothelial cells are crucial for the survival and self-renewal of these stem cells.

Markers such as ALDH, CD133, and CD44 have been effectively used to identify highly tumorigenic CSCs in HNSCC. Recent evidence indicates that CSCs are highly resistant to conventional therapies and act as the primary drivers of local recurrence and metastatic spread [125]. The CSC model highlights a tumor’s hierarchical structure, with self-renewing stem cells driving tumor growth and therapy resistance. Dysregulated signaling pathways like PI3K/Akt/mTOR, JAK/STAT, and Wnt play pivotal roles in CSC maintenance and therapy response. Inhibition of these pathways, such as with XAV939-targeting β-catenin in HNSCC and colon cancer, presents promising anti-CSC therapeutic strategies. Additionally, compensatory mechanisms like Ephs, TRKs, and c-Kit signaling sustain the CSC population in PIK3CA-overexpressing HNSCC, suggesting potential co-targeting opportunities. CSCs, located at invasive fronts and perivascular niches, are highly resistant to therapy, driving recurrence and metastasis.

### 3.4. Epithelial Mesenchymal Transition (EMT)

EMT is a physiologic cellular program that plays an important role during embryogenesis. In EMT, epithelial cells exhibit (1) morphological changes, from a cobblestone-like appearance to spindle-shaped cells; (2) differentiation marker changes, from cytokeratin intermediate to vimentin filaments; and (3) functional changes, from stationary to motile cells that can invade through the extracellular matrix [126]. Distinct cell populations with marked CSCs and EMT characteristics had been reported in OSCC. In primary OSCC tumors and cell lines, Biddle and colleagues demonstrated two distinct CSC populations: (1) CD44^high^EpCAM^high^ and (2) CD44^high^EpCAM^low^. Even though both populations were CSCs, as they formed tumors in NOD scid gamma severe combined immunodeficiency (NOD-SCID) mice, the CD44^high^EpCAM^high^ population was designated as non-EMT-CSCs, as this population exhibited more epithelial characteristics like cobblestone colony (holoclones) formation, more proliferative, and a lower rate of migration, with expression of epithelial-specific genes like E-cadherin and Keratin 15, while the CD44^high^EpCAM^low^ was designated as EMT-CSCs, as this population showed an elongated, fibroblast-like feature in a monolayer culture and a low proliferative rate but higher migratory ability, with overexpression of EMT markers like vimentin, Twist, Snail, and AXL. Furthermore, the study demonstrated that the cells of one population were able to generate cells of the other through EMT and mesenchymal epithelial transition (MET) processes and further identified a population with a MET ability restricted to another subpopulation, the CD44^high^EpCAM^low/−^ALDH^+^ cells [50]. In a separate study, the group reported a new CSC marker, CD24, which identified a drug-resistant EMT CSC subpopulation, CD44^high^EpCAM^low/−^CD24^+^ cells [56]. Other studies have also consistently documented such phenotypic plasticity in CSCs [59,65,127], raising questions on CSC heterogeneity and limitations in the methodologies for studying CSCs [24] (which will be discussed further in Section 4).

## 4. Cancer Stem Cells in Therapy

Resistance induced by the cancer cell remains a significant impediment to achieving effective therapeutic outcomes in HNSCC [128]. Despite the long-standing use of standard of care agents, such as cisplatin, their efficacy is limited, and they often lead to the development of chemoresistance [128]. Therefore, elucidating the molecular pathways that drive chemoresistance and targeting these pathways to re-sensitize tumors may be critical for advancing treatment strategies in HNSCC. While chemotherapeutic agents like cisplatin and 5-FU have been clinically approved for HNSCC, their therapeutic efficacy in metastatic and recurrent malignancy is markedly diminished due to acquired resistance [128]. Multiple factors contribute to this chemoresistance, with CSCs being a pivotal element. CSCs, characterized by their inherent self-renewal and plasticity, foster a more aggressive phenotype that is resistant to standard of care regimens. This CSCs subpopulation plays a critical role in immunosuppression, therapeutic resistance, metastasis, and invasion, ultimately leading to poor patient survival [34]. Addressing CSC-induced resistance through a deeper understanding of the underlying mechanisms and by targeting key pathways involved in chemoresistance holds promise for enhancing the efficacy of current therapeutic modalities and improving clinical outcomes in HNSCC [129]. Thus, Table 2 presents the current evidence on how CSCs and their downstream signaling are associated with chemoresistance, while also illustrating potential novel drug candidates to overcome CSC-induced chemoresistance in HNSCC preclinical models. Overall, approximately more than 80% of studies have exposed CSCs to a cisplatin regimen and characterized their resistance ability towards this chemotherapy due to the over-expression of CSC markers, such as β-catenin, polycomb complex protein BMI-1 (BMI1), ATP-binding cassette super-family G member 2 (ABCG2), CD44, ALDH1, Snail, CD10, Nanog, sex-determining region Y-box 2 (Sox2), octamer-binding transcription factor 4 (Oct4), CD133, neurogenic locus notch homolog protein 1 (Notch1), CD24, MYC oncogene (c-Myc), Krüppel-like factor 4 (Klf4), cyclin D1 (CCND1), forkhead box protein M1 (FOXM1), AlkB homolog 5, RNA demethylase (AKLBH5), CD271, CD326, and DEAD-box helicase family member (DDX3).

In 2013, Masui et al. [130] demonstrated that Snail-expressing HNSCC cell lines induced a stem-cell-like phenotype by increasing the expression of CD44 and ALDH1 and better resisted a cisplatin regimen as compared to non-Snail-expressing cell lines. Similarly, in 2015, Ota et al. [131] demonstrated a similar pattern observation that Snail expression enhances the chemoresistance to cisplatin in HNSCC cell lines. Khammanivong et al. [132] demonstrated that sphere cells with enrichment of CD44 and BMI1 CSC marker were highly resistant to cisplatin compared to monolayer cultures. Byun et al. [133] demonstrated that CD44^+^ HNSCC cells were relatively resistant to cisplatin and radiation, and knockdown of hypoxia-inducible factor (HIF-1) or Notch1 enhanced the chemosensitivity to cisplatin or radiation treatment as compared to the control. Lee et al. [134] demonstrated that Notch intracellular domain (NICD) activation enhanced sphere formation and also increased the expression of CD44, Oct4, and Sox2 markers. Knockdown of Notch1 enhanced the chemosensitivity to cisplatin treatment in HNSCC preclinical models. Xie et al. [135] reported that Sox8 enhances chemoresistance and upregulates CSC markers, while Sox8 knockdown increases chemosensitivity to cisplatin both in vitro and in vivo by reducing CD24^+^CD44^+^ CSC markers and tumor volume.

In Table 2, it can be seen that a few studies have utilized hyaluronic acid (HA, anti-CD44), inhibitors of apoptosis proteins (IAP), secreted frizzled-related protein 4 (sFRP4), curcumin, valproic acid (VPA), NCT-501, heat shock protein 90 (Hsp90) inhibitor, SB203580, XAV-939, SVC112, ketorolac, nuclear factor kappa-light-chain-enhancer of activated B cells (NF-κB) inhibitor, and TVB-3166 to overcome CSCs chemoresistance. Bourguignon et al. [136] and Ref. [137] demonstrated both HA treatment of tumor cells or CD44v3highALDH1 high cells promote sphere formation and confer chemoresistance to cisplatin treatment. To overcome the cisplatin resistant, a combination of SM164 [136] or anti-CD44 with cisplatin enhanced the cell-killing activity by targeting CSCs in HNSCC. Similarly, another approach from Warrier et al.’s [138] study demonstrated that a combination of sFRP4 and cisplatin exhibited the lowest cell viability and highest apoptotic cell death as compared to monotherapy of sFRP4, cisplatin, and an untreated control in a cisplatin-resistance preclinical model. Apart from that, Basak et al. [139] demonstrated that CD44^hi^ cells exhibit properties of CSC that confer cisplatin resistance. A combination of curcumin or curcumin-difluorinated (CDF), a synthetic analog of curcumin, with cisplatin increased the cell number reduction in CD44^high^ CSCs’ cell population that confer cisplatin resistance. Using drug repurposing approaches, Lee et al. [140] demonstrated that a combination of VPA, a drug used for complex partial seizures, with cisplatin overcomes cisplatin resistance driven by CSCs in an HNSCC preclinical model. Kulsum et al. [141] focused on targeting ALDH1A1 in CSCs that promote chemoresistance to cisplatin treatment using NCT-501. Post-treatment with this inhibitor overcame chemoresistance by reducing the spheroid formation in vitro and reduced the tumor volume in vivo as compared to the untreated control. A similar trend was also observed in Subramanian et al.’s [142] study, which employed the Hsp90 inhibitor to reduce both CD44 and ALDH subpopulations in vitro and reduced tumor volume in vivo, which were resistant to cisplatin treatment. Few studies have been conducted by the Banerjee lab at Kalinga Institute of Industrial Technology using various inhibitors to target CSC markers in HNSCC to enhance chemosensitivity in preclinical models resistant to cisplatin [118,143,144,145]. For instance, Roy et al. [143,144] employed a mitogen-activated protein kinase (p38) inhibitor to target CSC markers in HNSCC, resulting in a decrease in the half maximal inhibitory concentration (IC50) value of cisplatin compared to untreated controls resistant to cisplatin treatment. They also demonstrated that the combination of XAV-939, a tankyrase inhibitor targeting β-catenin, with cisplatin synergistically attenuated chemoresistance by increasing DNA damage in vitro [118]. In 2020, Roy et al. [145] demonstrated that tetrahydroisoquinoline (THIQ)-mediated inhibition of CD44 sensitized HNSCC cells to cisplatin treatment. Similarly, Keysar et al. [146] demonstrated that SVC112, a translation elongation inhibitor, enhanced tumor reduction in vivo when combined with cisplatin and radiation regimens. Additionally, Shriwas et al. [147] showed that the combination of ketorolac, a bioactive inhibitor of DDX, with cisplatin enhanced the chemosensitivity of cisplatin-resistant cells by reducing the IC50 of cisplatin in vitro and inhibiting tumor growth in vivo. Recently, de Castro et al. [68] and Aquino et al. [66] demonstrated that the combination of an NF-κB inhibitor and a selective fatty acid synthase (FASN) inhibitor, respectively, enhanced the chemosensitivity of cisplatin in an in vitro model resistant to cisplatin treatment in HNSCC.

In Table 2, we also identify two novel drugs, namely, casein kinase 2 (CK2) inhibitor and MEDI0641, which were found to suppress CSC subpopulations in HNSCC. Lu et al. [148] demonstrated the CK2 inhibitor suppressed CSC markers expression, such as *Nanog*, *Oct4*, and *Sox2* mRNA and protein expression, as well as the CSC side population phenotype. Similarly, Kerk et al. [149] demonstrated the ability of using an antibody-drug conjugate (ADC) to target the 5T4 oncofetal antigen known as MEDI0641 and reduce CD44 and ALDH cell populations, as well as reduce tumor growth and prevent tumor recurrence in an in vivo model of HNSCC.

5-FU is a commonly used chemotherapeutic agent for HNSCC patients, often employed alongside cisplatin or as a standalone therapy. Tabor et al. [150] demonstrated that the side population, a group of cells exhibiting stem-cell-like characteristics, showed resistance to 5-FU treatment compared to the non-side population cells. Other studies have shown that HNSCC cell lines exhibiting a CSC phenotype confer chemoresistance to monotherapy with cisplatin, 5-FU, or radiotherapy. For instance, Elkashty et al. [61] and Fukusumi et al. [151] demonstrated that the CD44^+^CD271^+^ and CD10 populations, respectively, exhibited a CSC phenotype and conferred resistance to monotherapy with cisplatin, 5-FU, or radiotherapy in vitro.

**Table 2 biomedicines-12-02111-t002:** Summary of findings on chemoresistance mechanisms and potential therapeutic strategies targeting CSCs in HNSCC.

References	Marker(s)	Cell line(s)	Drug(s)	Findings
[150]	β-catenin BMI1 ABCG2	UM-SCC-10B	5-Fluorouracil (5-FU) (0–100 μM)	The side population cells were characterized with high expressions of BMI-1 and ABCG2 and a low expression of β-catenin and exhibited greater resistance to the chemotherapeutic agent 5-FU. BMI-1 suppresses p16Ink4a expression and enhances self-renewal abilities, whereas ABCG2 confers drug resistance through the efflux of chemotherapeutic drugs. Additionally, the decrease in β-catenin levels inhibits its translocation to the nucleus, thereby shielding cells from the cytotoxic impact of 5-FU.
[136]	CD44v3 ALDH1	HSC-3	HA (50 μg/mL) (or anti-CD44 antibody plus HA (50 μg/mL) or no HA) Cisplatin (4 × 10−9 to 1.75 × 10−5 M) IAP inhibitor (SM-164)	The interaction between HA and CD44v3 with Oct4-Sox2-Nanog signaling stimulates the production of miR-302, which subsequently results in the downregulation of AOF1/AOF2/DNMT1. This process enhances the survival and resistance to chemotherapy in tumor cell populations characterized by high levels of CD44v3 and ALDH1. Furthermore, the concurrent use of SM164 and cisplatin leads to increased efficacy in killing cancer cells by specifically targeting cancer stem cells in HNSCC.
[130]	Snail CD44 ALDH1	SAS HSC-4	Cisplatin (1–10 μM)	Induction of Snail increases expression of CD44 and ALDH1. The induction of EMT via Snail suppresses E-cadherin expression that leads HNSCC cells to adopt CSC-like phenotype and chemoresistance to cisplatin. Focusing on the strategic targeting of EMT-regulating Snail presents a potential benefit in cancer therapy, as suppressing EMT could effectively impede cancer progression and spread, while also preventing the development of cancer stem cells.
[151]	CD10	FaDu Detroit562	Cisplatin (0.1–5 μM) 5-FU (0.5–50 μM)	A CD10^+^ positive subpopulation of HNSCC cells exhibited CSC-related properties, such as chemo- and radioresistance, a self-renewal capacity, and enhanced tumorigenicity. CD10 has been shown to be mechanistically associated with the increased expression of OCT3/4, a pivotal regulator of both stem cell self-renewal and oncogenic pathways. Silencing CD10 led to a reduction in OCT3/4 levels, implying that CD10 promotes cancer stem cell characteristics in HNSCC by enhancing OCT3/4 expression. These results indicate that targeting CD10 could potentially be a beneficial strategy for addressing the challenges of chemoresistance and radioresistance in treatment-resistant HNSCC.
[138]	CD44^+^ ALDH^+^	Hep2 KB	sFRP4 (Wnt antagonist, 250 pg/mL) Cisplatin (10 Mm)	A decrease in the expression of *ABCG2* and *ABCC4* was observed in sFRP4-drug-treated spheroids. A combination of sFRP4 and cisplatin exhibits the lowest cell viability and highest apoptotic cell death as compared to monotherapy of sFRP4, cisplatin, and an untreated control.
[148]	Nanog Sox2 Oct4	UM-SCC-1 UM-SCC-46	CX-4945 (CK2 inhibitor, 0.5, 1 and 5 μM)	Inhibition of CK2 and activation of TAp73 repress the mRNA and protein expression of Nanog, Oct4, and Sox2, along with reducing the CSC side population phenotype. This process promotes tumor protein P73 (TAp73) expression, leading to growth arrest and the overexpression of pro-apoptotic genes like cyclin-dependent kinase inhibitor 1 (p21WAF1) and the p53 upregulated modulator of apoptosis (PUMA).
[132]	CD44 BMI1	TR146 SCC-58 UMSCC-17B	Cisplatin (0 to 10 μM)	Spheroids grown on Matrigel exhibited elevated expression of CD44 and BMI1, with a higher proportion of cells in the CD44^high^ population, both of which are markers for HNSCC CSCs. These spheroids demonstrated significantly higher resistance to cisplatin compared to monolayer cultures, with the ALDH activity remaining either unchanged or slightly reduced in the spheroid cells. The increased expression of SMAD-specific E3 ubiquitin protein ligase 1 (SMURF1) in the spheroid cultures results in the suppression of traditional BMP signaling, specifically, the decreased activation of pSMAD1/5/8, which ultimately enhances resistance to chemotherapy. To combat this, addressing SMURF1 levels and reinstating BMP signaling could present a novel treatment strategy to stimulate differentiation and decrease the population of cancer stem cells, ultimately diminishing drug resistance and the likelihood of disease recurrence.
[139]	CD44 CD133 ALDH	CCL-23 UM-SCC-1	Curcumin (25 μM) Cisplatin (10–20 μM)	In vitro models resistant to cisplatin demonstrated heightened levels of IL-1β, IL-6, IL-8, IL-10, basic fibroblast growth factor (bFGF), vascular endothelial growth factor (VEGF), matrix metalloproteinase-1 (MMP-1), and matrix metalloproteinase-9 (MMP-9), which were associated with the activation of the AKT pathway (GSK-3β, p70S6K), thereby leading to the development of chemoresistance. The in vitro and in vivo inhibitory effect on CD44 clearly demonstrates that curcumin targets CSCs and could reduce CD44^hi^ CSC accumulation with cisplatin treatment.
[140]	Sox2 Oct4	K3 K5	Cisplatin (5 μM) VPA (400 μM)	VPA disrupted the self-renewal of HNSCC CSCs by downregulating the expression of stem cell markers, such as Oct4, Sox2, and CD44. Combining VPA with cisplatin further diminished HNSCC CSC chemoresistance, likely by suppressing ABCC2 and ABCC6 expression and enhancing caspase-mediated apoptosis.
[131]	Snail	SAS HSC-4	Cisplatin (1 μM)	Snail overexpression induced CSC-like properties, and the expression of phospho-EGFR was enhanced in the Snail-transfected cells compared to the controls.
[134]	Notch1	SNU 1041	Cisplatin (0–50 μM)	Activation of Notch1 signaling is correlated with increased cell proliferation in HNSCC through its influence on cell cycle progression, while constitutive activation of NICD promotes CSC traits in differentiated HNSCC cells. Reducing Notch1 signaling attenuates CSC traits in HNSCC-derived CSCs and enhances their chemosensitivity to cisplatin by suppressing the expression of ABCC2 and ABCG2 transporter genes.
[137]	CD44v3^high^ ALDH1^high^	HSC-3	Anti-CD44 antibody/HA (4 × 10^−9^ to 1.75 × 10^−5^ M) Cisplatin (4 × 10^−9^ to 1.75 × 10^−5^ M)	HA treatment promotes sphere formation, self-renewal/growth, and differentiation, such as tumor cell invasion, in highly tumorigenic CD44v3^high^ALDH1^high^ tumor cells, indicating HA signaling’s role in regulating CSC properties.
[141]	CD44 ALDH1A1	Cal27 Hep-2	NCT-501 (ALDH1A1 inhibitor, 0–80 nM) Cisplatin (20 μM)	CSC-specific markers (CD44, ALDH1A1, CD133, Notch1), a drug efflux marker (ABCG2), and stem cell maintenance markers (Sox2, Nanog, Oct4) showed differential upregulation in resistant cell lines, correlating with increased spheroid formation and migration. Silencing ALDH1A1 in cisplatin-resistant cell lines led to decreased multidrug resistance (MDR) gene expression, downregulation of CD44 and CD133, loss of self-renewal and migratory properties, and increased cisplatin sensitivity. Post-treatment with NCT-501 reduced the tumor volume significantly as compared to the untreated control in an in vivo model.
[149]	ALDH^high^ CD44^high^	UM-SCC-11B UM-SCC-22B	(MEDI0641, Antibody-drug conjugate targets 5T4 oncofetal antigen, 0–1 μg/mL)	A single dose of MEDI0641 led to long-term tumor regression in patient-derived xenograft (PDX) models of HNSCC, even after tumors were already established. MEDI0641 effectively targets and reduces the fraction of CSCs, which express high levels of 5T4, in both HNSCC cell lines in vitro and in vivo.
[142]	CD44 ALDH	UMSCC- 22B MDA1986	Hsp90 inhibitor KU711 (20 to 40 μM) KU757 (1.0 to 2.5 μM) Cisplatin (2μM)	Both Hsp90 inhibitors reduced sphere formation in vitro by decreasing the CD44 and ALDH populations, which were resistant to cisplatin treatment. Additionally, these inhibitors led to a reduction in tumor volume in vivo in cisplatin-resistant models.
[135]	Sox2 Oct4 BMI1 ABCG2 CD44 CD24	Cal27 SCC9	Cisplatin (0^−7^ M to 10^−5^ M)	Ectopic expression of Sox8 promotes chemoresistance, CSC properties, and EMT features in chemoresistant HNSCC cells. Knockdown of Sox8 inhibited chemoresistance, CSC properties, and EMT features. Post-treatment with cisplatin reduced the tumor volume in xenograft models with Sox8-knockdown cell lines compared to cisplatin alone or untreated controls.
[143]	c-Myc CD44 Oct4 Klf4	UPCI-SCC-131 Cal27	SB 203580 (p38 inhibitor, 10 μM) Cisplatin (1–20 μM)	p38-inhibited cells exhibit a lower IC50 of cisplatin and reduced expression of CSC markers and increased apoptosis as compared to the untreated control.
[118]	c-Myc CD44 Oct4 Klf4	UPCI-SCC-131 Cal27	Cisplatin (1–20 μM) XAV-939 (Tankyrase inhibitor, 1–4 μM)	The combination of cisplatin and XAV-939 significantly reduced the expression of the CSC marker Oct4, the nucleotide excision repair gene excision repair cross-complementing rodent repair deficiency, complementation group (ERCC1), and β-catenin. This combined treatment also heightened the genotoxic effects, as indicated by increased DNA damage and shortened telomere length. Together, cisplatin and XAV-939 contribute to genomic instability in HNSCC cells by amplifying DNA damage.
[146]	c-Myc CCND1 CD44 Sox2	036C 067C 013C 049C Fadu Detroit562	SVC112 (0–1000 nM) Cisplatin (1 mg/kg once weekly)	SVC112, an elongation inhibitor, significantly suppressed the expression of c-Myc, CCND1, and Sox2 in HNSCC cells and CSCs across in vitro, ex vivo, and in vivo models. When combined with cisplatin and radiation therapy, SVC112 further reduced tumor growth in vivo.
[147]	Sox2 Oct4 FOXM1 Nanog ALKBH5	H357 SCC-9 SCC-4	Cisplatin (0–10 μM/2.5 mg/kg) Ketorolac (0–2.5 μM/30 mg/kg)	A combination of cisplatin with ketorolac enhanced the chemosensitivity of cisplatin in cisplatin-resistant cell lines in in vitro and in vivo models by targeting the DDX3 and CSC subpopulations.
[145]	CD44	UPCI-SCC-131 Cal27	1,2,3,4 tetrahydroisoquinoline (THIQ) (1–5 mM) Cisplatin (1–10 μM)	The impact of CD44 inhibition on the expression of proteins involved in the Wnt/β-catenin signaling pathway, as well as on other CSC markers like Oct4 and Klf4, was examined. Inhibiting CD44 led to reduced levels of β-catenin and phosphorylation of GSK3β (Ser 9), alongside decreased expression of other CSC markers. Treatment with 1 mM THIQ resulted in a significant reduction in CD44 expression in both cisplatin-resistant HNSCC cells and their parental counterparts. Additionally, a flow cytometric analysis revealed that 1 mM THIQ treatment decreased the percentage of CD44-positive cells.
[61]	CD44^+^/CD271^−^ CD44^+^/CD271^+^	SCC38 SCC12	Cisplatin (2 µg/mL) 5-FU (32 µg/mL)	CD44^+^/CD271^+^ cells exhibited superior colony-forming and sphere-forming abilities compared to CD271^−^ cells and their unsorted parental counterparts, and they also formed larger spheres. This subpopulation of CD44^+^CD271^+^ cells demonstrated the highest resistance to cisplatin, 5-FU, and radiation regimens when compared to CD44^+^CD271^−^ cells and parental cell line models. The potential explanation for this phenomenon could stem from the ALDH1A1-mediated stimulation of the ATP-binding cassette subfamily B member 1 (ABCB1) drug-efflux pump, as well as survival proteins, such as AKT and BCL2. Focusing on the CD271 cell subset results in the ability to overcome chemotherapy resistance in HNSCC.
[144]	c-Myc CD44 Oct4 Klf4	SCC-131 Cal 27	SB 203580 (p38 inhibitor, 10 μM) Cisplatin (1–10 μM)	Treatment with escalating doses of cisplatin resulted in an increased expression of CSC markers. In contrast, inhibition of p38 in cisplatin-resistant cells led to a reduced expression of CSC markers, as the cisplatin concentration increased. Additionally, p38 inhibition reduced both the size and sphere-forming ability of cisplatin-resistant cells compared to cells without p38 inhibition, suggesting that p38 plays a role in maintaining the CSC phenotype in HNSCC cells.
[133]	CD44^+^	SCC-15 SCC-25	Cisplatin (5 μM)	CD44^+^ HNSCC cells exhibited relative resistance to cisplatin and radiation. However, knockdown of HIF-1α or Notch1 increased chemosensitivity to cisplatin and radiation compared to the control cells.
[68]	ALDH^high^ CD44^high^	Cal 27 SCC9	Cisplatin (5 Μm) NF-κB inhibitor CBL0137 (0–10 μM) Emetine (0–10 μM)	In CisR cell lines, overexpression and phosphorylation of NF-κB were observed. Treatment with NF-κB inhibitors reduced sphere formation and decreased the expression of *NF-κB*, chemokine (C-X-C motif) ligand 8 (*CXCL8)*, and *TNF-alpha* mRNA. Combining cisplatin with NF-κB inhibitors further reduced the ALDH^high^/CD44^high^ populations and the number of colonies formed compared to both the untreated control and cisplatin monotherapy.
[66]	CD44 CD326	LN-1A	TVB-3166 (0–150 μM) Cisplatin (0–100 μM)	Three subpopulations known as CD44^Low^/CD326^−^ (CSC-M1), CD44^Low^/CD326^High^ (CSC-E), and CD44^High^/CD326^−^ (CSC-M2) were isolated from LN-1A. Post-treatment with cisplatin showed that CSC-M1 has a lower IC50 compared to LN-1A, CSC-E, and CSC-M2, whereas both CSC-E and CSC-M2 have higher IC50 values compared to LN-1A, with CSC-M2 having the highest IC50. Exposure to TVB-3166 reduced the IC50 value for all three subpopulations compared to LN-1A.

The U.S. FDA has recently approved checkpoint inhibitors like pembrolizumab and nivolumab for recurrent metastatic HNSCC [152]. However, their response rate has been modest, at around 20% [153]. This indicates that intrinsic properties of the TME or specific cells within it contribute to resistance, driving recurrent and secondary tumor growth. A key factor in this resistance may be related to immune evasion mechanisms, where tumors downregulate peptide major histocompatibility complex molecules and employ other strategies to avoid immune detection [154]. CSCs are likely to play a significant role in this process, making it crucial to understand their resistance not only to standard therapies like chemotherapy and radiation but also to novel treatments, such as immune checkpoint inhibitors [155]. Gaining this knowledge can provide valuable insights into how we might target CSCs, re-sensitize immune cells, and ultimately enhance the therapeutic efficacy of checkpoint inhibitors, not only for HNSCC but also for other types of cancers.

The current therapeutic achievements of HNSCC have been very challenging due to the CSC hypothesis. A major reason behind this challenge is the role of CSCs to metastasize into distant organs and be resistant to therapies. One major component that drives a successful metastasis is the ability to invade the immune system [156]. The development and introduction of immunotherapy in HNSCC holds very promising as a supplement to the traditional standard of care, like surgery, chemotherapy, and radiation; however, the effect of this therapy has not reached its peak in the realm of HNSCC [157]. A major reason behind this setback is the ability of CSCs to recognize immune cells and mount an immunosuppressive environment within the TME and during its transit of metastasis [158]. Therefore, it is crucial to understand how CSCs drive the potency to suppress the immune system and how this interaction can be leveraged to gain further insights in developing novel therapies that can help in achieving better therapeutic success for patients diagnosed with HNSCC. It has been shown that the expression of CSC markers in patient tumor tissue is dependent on the number of tumor-infiltrating immune cells. The most important interaction between the T cell receptor on cytotoxic T cell lymphocytes (CTL) with the major histocompatibility complex (MHC) class I antigen is the one that drives the tumor cell recognition and cytotoxic phenomenon by T cells; however, it is suggested that the steps of this process of antigen processing and presenting are impaired in CSCs [159]. In accordance with the previous statement, recent studies have shown MHC downregulation in melanoma spheroid cells, leading to inhibition of the allogenic immune response of T cells. Similarly, defects in the MHC class I directed antigen presentation to CSC have no cytotoxic effect. Moreover, it has been studied that this phenomenon is not just restricted to CSCs but is also present in normal hematopoietic cells and epithelial cells that express Lgr5^+^, which is a stem cell marker in intestinal crypts and mammary glands and fails to induce antigen presentation [160,161,162]. It was also demonstrated that CSCs have been involved in the downregulation of the MHC molecule in HNSCC. HNSCC cells with CD44^+^ exhibit stem cell properties that have shown the downregulation of many common human leukocyte antigen (HLA) classes, like HLA-A2, HLA class II [163]. Furthermore, CD44^+^ HNSCC has been known to induce the immunosuppressive cells, like regulatory T cell (Treg) and myeloid-derived suppressor cells (MDSC), which have high immunosuppressive cytokines like interleukin-8 (IL-8) and transforming growth factor beta [158,163]. The expression of the immune checkpoint is well known by now to induce immunosuppression; however, its role in CSCs has not yet been conclusive enough. However, a recent study suggests that there can be other possible mechanisms that help CSCs to escape immune surveillance. CD276, which is a member of the B7 family of immune checkpoint proteins and is highly expressed in tumor cells, has also been seen to be expressed in CSCs in mouse HNSCC. When treating with anti-CD276 antibodies, it was observed that CSCs were eliminated by CD8^+^ T cells, which also inhibited tumor growth and nodal metastasis. Furthermore, RNA sequencing data showed the blockade of CD276 remodels SCC heterogeneity and reduced the EMT transition that is a crucial step in inducing metastasis [164,165].

HNSCC is characterized by its cold TME, which means the lack of efficient and sufficient tumor-infiltrating lymphocytes, which fosters a favorable microenvironment for CSCs to evade and metastasize. A recent study delved deeper into the aspect in the context of HNSCC, where they have characterized the presence of CD34^+^ cells producing a high level of IL-6, which participates in the development of TME and triggers its own malignant progression [166]. Other immune checkpoints that have played significant roles are CD276, leukocyte immunoglobulin-like receptor subfamily B member 2 (LILRB2), and CD47, which were upregulated in CSCs and led to a weakened or damaged host anti-tumor response. The presence of naïve CSCs appears to be more malignant and results in a worse prognosis [167]. The therapeutic landscape for HNSCC faces a significant challenge due to CSCs. CSCs can evade immune surveillance through different mechanisms, including downregulation of MHC and induction of immunosuppressive cells and cytokines. Recent studies suggest that targeting immune checkpoints like CD276 in CSCs may enhance anti-tumor responses and inhibit metastasis. Additionally, the cold TME in HNSCC further facilitates CSC evasion and metastasis, highlighting the importance of understanding and targeting these interactions for improved therapeutic outcomes.

## 5. Challenges and Future Directions

As evidenced in Table 1, the lack of specific CSCs markers is a challenge in the study of CSCs. The choice of markers is usually based on previous reports that have successfully demonstrated CSCs with a particular marker. For example, CD44 was first used as a marker to study CSCs in HNSCC because of its success in identifying breast CSCs [44]. Recently, by using previously reported putative CSCs markers, including CD44, CD24, and ALDH, Vipparthi and colleagues demonstrated the presence of four CSC subpopulations (CD44^+^CD24^low^ALDH^high^, CD44^+^CD24^low^ALDH^low^, CD44^+^CD24^high^ALDH^high^, CD44^+^CD24^high^ALDH^low^) in OSCC. All these subpopulations exhibited CSC traits by forming spheres in suspension culture. Moreover, self-renewal and differentiation were observed in these subpopulations; for instance, upon repassage, the CD44^+^CD24^low^ALDH^high^ subpopulation gave rise to itself and the other three subpopulations, which overexpressed genes of differentiation. Additionally, the subpopulations also exhibited phenotypic plasticity upon exposure to cisplatin. At sub-lethal doses, the CD44^+^CD24^high^ALDH^low^ cell phenotype was induced into the CD44^+^CD24^high^ALDH^high^ cell phenotype; the group further attributed the phenotypic change to overexpression of ABCG2 and Sox9 genes rather than apoptosis-mediated selection [168].

Hence, all together, these experimental characteristics of CSCs have indicated that (1) a CSC population is heterogeneous [167] and (2) any given CSC marker can only isolate a fraction of all CSCs [24]. Without specific markers, targeting CSCs for therapeutic elimination will be difficult [169]. As such, other experimental approaches should be looked into to better study CSCs. Here, we propose the use of 3D cell culture technologies to advance the study of CSCs in HN-/OSCC.

### Three-Dimensional Cell Culture System in OSCC

Initially used in a neural stem cell study, a sphere formation assay is increasingly being used in CSC studies. In OSCC, CSCs identified and isolated by putative CSC markers consistently proliferated as free-floating spheres in suspension culture (refer to Table 1). Some authors have named these spheres orospheres [87]. As such, the development of non-adherent culture systems has provided an alternative method to study CSCs in addition to the aforementioned marker-based approach. Using OSCC cell lines, Chen and colleagues developed a non-adherent culture system to isolate OSCC-CSCs and reported that spheres generated with this method retained all the CSC traits, as they formed tumors more efficiently when injected into BALB/c nude mice, overexpressed CD133 and ALDH1, and were more chemoresistant than the parental cells [85]. Similarly, sorted CD44^+^ALDH^+^ HNSCC cancer cells cultured in ultralow attachment and soft-agar cultures grew as free-floating spheres, which can be propagated with little loss of CSC traits upon several passages [87]. To address the problem of marker specificity, Pozzi and colleagues demonstrated that a sphere formation assay could reliably identify and enrich CSCs from HNSCC for better characterization [55].

Spheres, orospheres, spheroids, multicellular tumor spheroids (MCTS), tumoroids, and organoids are terms used to name proliferating cancer cells grown as 3D cellular structures. Essentially, in these systems, cancer cells are first dissociated into a single-cell suspension and are then grown in scaffold-free or scaffold-based techniques. In a scaffold-free culture, cancer cells freely grow as 3D aggregates. In the scaffold-based technique, however, hydrogel-based support, which mimics the extracellular matrix, is used to support three-dimensional cell growth. Cancer cells grown as 3D structures better replicate the three-dimensional in situ complex characteristics of a tumor. The scaffold-based techniques also provide a platform for modification of the culture system by modifying the types of hydrogel-based supports to more closely replicate the in vivo TME [170,171].

As such, 3D cell culture methods are increasingly being used in HN-/OSCC. After successfully generating organoids derived from healthy oral mucosa, Driehuis and colleagues generated tumoroids from 31 fresh primary HNSCC samples with the use of basement membrane extract (BME) in the scaffold-based technique. The cancer cells grew as tumoroids in this culture system without contamination from immune, connective, or vascular tissues or normal epithelium cells. In addition, the tumoroids also recapitulated original genetic alterations in vitro and in vivo when xenografted. Furthermore, a biobank of tumor organoids was established and used as a tool for treatment outcome prediction and therapeutic exploration. The group reported that the organoids’ response to irradiation mimicked the patients’ response clinically when irradiated with a fraction of a dose corresponding to the treatment dose. The group also discovered that cyclin-dependent kinase inhibitor 2A (CDKN2A) null organoids demonstrated increased sensitivity to EZP01556, a protein arginine methyltransferase 5 (PRMT5) inhibitor, indicating a potential new treatment option for many HNSCC patients who harbored a loss of CDKN2A [172,173]. In another study using the scaffold-free technique, MCTSs derived from HN cancer in 384-well U-bottom, ultralow attachment microplates exhibited permeability barriers that are frequently seen in patient tumors. Fluorescent images of the MCTSs after exposure to chemotherapeutic agents demonstrated a similar pattern of drug penetration in patient tumors and mouse xenografts; the MCTSs exposed to chemoreagents showed penetration of the drugs in the cells of the outer layer of MCTSs, resulting in uneven drug distribution within the MCTSs. This ability to recapitulate drug penetration indicates MCTSs mimic physiological TME and can be used as a model for drug testing [174,175].

A similar approach has also been used in the study of CSCs. A stem-cell-enriched spheroid model (SCESM) was developed [176]. In comparison to adherent cells, the tumor spheroids thus generated showed an increased expression of stem markers, such as ALDH1A1, CD44, and CD133, and core transcription factors of the human stem cell pluripotency signaling pathway, such as *Oct4*, *Sox2,* and *Nanog*. It was also shown that the tumor spheroids were composed of fast proliferating cells on their outer rims and slow or non-proliferating cells at their cores. This is in corroboration with a metabolomic study on CSCs using OSCC-MCTSs, which reported that CSCs relied more on glycolytic than oxidative phosphorylation, with a reduction in fatty acid oxidation activity, suggesting that the CSCs were weakly proliferative [177]. Additionally, a histomorphological analysis of spheroids derived from OSCC cell lines demonstrated a consistent histomorphological pattern of OSCC with a central zone composed of highly pleomorphic and polyhedral cancer cells and more cohesive, flatter cancer cells on the outer rim [178].

One of the common uses for 3D cell culture models has been for drug screening and discovery. In a study using organoids generated from an HNSCC patient, Tanaka and colleagues demonstrated that the resistance displayed by these organoids to cisplatin and docetaxel coincided with recurrence in the patient receiving prior treatment with these chemo reagents [179]. In another study assessing the utility of spheroids from OSCC cell lines to cisplatin and cetuximab, Ono and colleagues reported that cancer cells grown as 3D structures better reflected drug responsiveness than a monolayer culture [114]. Using the 3D model, Zhao and colleagues further demonstrated the potential of targeting the lactate uptake pathway with monocarboxylate transporter 1 (MCT1) knockout in ablating OSCC-CSCs [180].

As cell culture technology continues to evolve, more advanced 3D cell culture models are developed. Ikeda-Motonakano and colleagues developed a high-throughput microfabricated microwell device, which accommodated the growth of 195 spheroids derived from OSCC cell lines, to study CSCs. The spheroids in this system displayed characteristics of CSCs with the formation of a tumor when xenografted, overexpressed CSCs markers such as CD44, and increased resistance to cisplatin, rendering this system useful for the screening of CSC-targeting drug candidates [181]. Recently, Chen and colleagues used a microfluidic chip to isolate a single cell from OSCC for CSC study. The group showed that the microfluidic-chip-assisted capture of OSCC cancer cells yielded significant sphere formation when cultured with the scaffold-free technique. These cells also exhibited CSC characteristics as previously discussed [182].

Taken together, as 3D cell culture technologies continue to evolve and incorporate other advancing technologies in tissue bioengineering, such as microfluidic organ-on-a-chip platform, these models will further replicate the physiological microenvironment more closely for the development of personalized medicine protocols for OSCC patients.

## Figures and Tables

**Figure 1 biomedicines-12-02111-f001:**
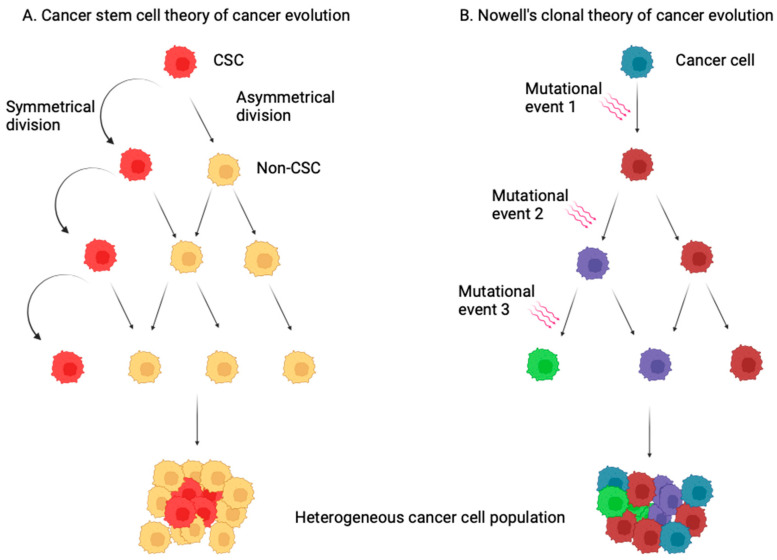
Evolution of cancer. (**A**) The cancer stem cell (CSC) theory of cancer evolution proposes the existence of a hierarchy in the heterogeneous cancer cell population in which cancer stem cells (CSCs) occupy the apex of the hierarchy. Through symmetrical and asymmetrical division in each cell cycle, CSCs give rise to other CSCs (through self-renewal) and non-CSCs, contributing to the heterogeneity in the cancer cell population. (**B**) In Nowell’s clonal theory of cancer evolution, all cancer cells are equally probable in acquiring more mutations, and each cell will expand clonally to form a heterogeneous cancer cell population.

**Figure 2 biomedicines-12-02111-f002:**
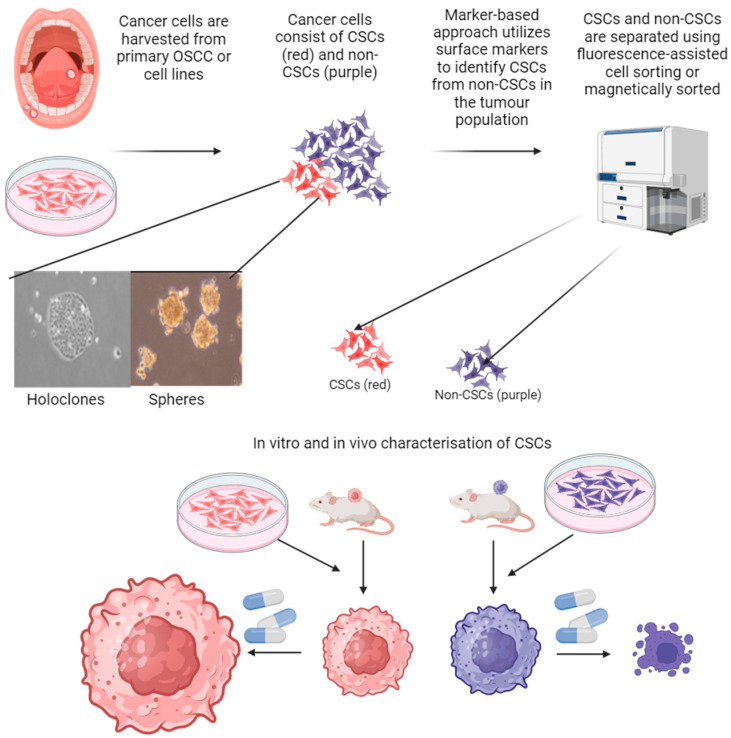
CSCs in OSCC. Cancer cells harvested from primary OSCC or cancer cell lines consist of a heterogeneous cancer population of CSCs and non-CSCs. CSCs are identified and then sorted by using cell surface markers. CD133 and CD44 are commonly used cell surface markers to identify and sort OSCC CSCs using fluorescence-assisted cell sorting (FASC) or magnetic beads in magnetic sorting for further characterization of CSCs. When cultured as a monolayer, CSCs grow as tightly packed cell colonies (holoclones). In non-adherent culture, CSCs readily form tumor spheres. Most importantly, CSCs form tumors rapidly when xenotransplanted into an animal model and resist chemotherapy regimens.

**Table 1 biomedicines-12-02111-t001:** Summary of in vivo evidence of CSCs in HNSCC.

References	Markers	Cell Line	Mice Strain	Site
[44]	CD44^+^	Primary HNSCC tumors	NOD/SCID or Rag2γDKO	S.C. (Right and left flank)
[45]	CD133^+^	Primary OSCC tumors or cell lines	BALB/c nude	S.C. (Back region)
[46]	CD44^+^CD24^−^ ALDH^+^	Primary HNSCC tumors	SCID	S.C. (Neck region)
[47]	ALDH^high^	Primary HNSCC tumors	NOD/SCID	S.C. (Site N/A)
[48]	CD44^+^ ALDH^+^	Primary HNSCC tumors	BALB/c nude	S.C. (Neck region)
[49]	CD133^+^	OSCC cell lines	Athymic NCr-nu/nu	S.C. (Right and left midabdominal area)
[50]	CD44^high^EpCAM^high^	OSCC cell lines	NOD/SCID	Orthotopic (Tongue)
[51]	CD44^+^CD133^+^	OSCC cell lines	BALB/C nude	S.C. (Roots of mouse limb)
[52]	CD24^+^CD44^+^	HNSCC cell lines	Athymic nude immunodeficient	S.C. (Dorsal flank)
[53]	Cisplatin-resistant cells	HNSCC cell lines	SCID	S.C. (Flank)
[54]	Side population	OSCC cell lines	BALB/c	S.C. (Right and left midabdominal area)
[55]	Spheres	HNSCC and OSCC cell lines	BALB/c nu/nu	S.C. (Flank)
[56]	CD44^high^/EpCAM^low^/CD24^+^	OSCC cell line	NOD/SCID	Orthotopic (Tongue)
[57]	CD44^+^CD66^−^	Primary HNSCC tumors	Rag-2/γc^−/−^	S.C. (Neck region)
[58]	CD44^+^ALDH1^+^	OSCC cell lines	BALB/c nu/nu	S.C. (Right front axilla)
[59]	CD44^+^CD24^low/−^	OSCC cell lines	Athymic NCr-nu/nu	Orthotopic (Tongue)
[60]	BMI1^+^	Primary 4NQO-induced HNSCC mouse model	C57BL/6, NOD/SCID	Orthotopic (Tongue)
[61]	CD44^+^CD271^−^	HNSCC cell lines	NU/NU nude (Crl:NU-Foxn1^nu^) mice	Orthotopic (Tongue)
[62]	CD133^+^	Primary OSCC tumors	BALB/c mice	S.C. (Exact site N/A)
[63]	CD44^+^CD133^+^	Primary OSCC tumors	BALB/c nude	S.C. (Exact site N/A)
[64]	CD44^high^EpCAM^high^	OSCC cell lines	NOD/SCID	Orthotopic (Tongue)
[65]	CD44^high^CD24^low^	OSCC cell lines	NOD/SCID mice	S.C. (Dorsal flank)
[66]	CD44^high^CD326^−^ CD44^low^CD326^high^ CD44^low^CD326^−^	OSCC cell lines	BALB/c nude	Orthotopic (Tongue)
[67]	CD44^+^	OSCC cell lines	BALB/c nude	S.C. (Forelimb)
[68]	Cisplatin-resistant cells	OSCC cell lines	BALB/c nude	S.C. (Right and left back region)

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
