# Peer review of "Cancer Stem Cells in Oral Squamous Cell Carcinoma: A Narrative Review on Experimental Characteristics and Methodological Challenges"

_biomedicines, 2024, doi:10.3390/biomedicines12092111_

Round 1

Reviewer 1 Report

Comments and Suggestions for Authors

Dear authors, I carefully read and revised the manuscript entitled “Strategies for targeting Cancer Stem Cells in Oral Squamous Cell Carcinoma: A Comprehensive Review”. The review article aims to resume the current knowledge about cancer stem cells in Oral Squamous Cell Carcinoma, and the possible strategies to avoid recurrence and metastasis. In my opinion, the manuscript is not focused on the topic proposed by the title: paragraphs are not always clearly discussed and sometimes the purpose of the work is lost. For these reasons, I reported some suggestions to improve the review.

1-     The review article lacks a methods paragraph in which it is possible to determine the search strategy used, and the number of studies evaluated or selected. I suggest the authors add a dedicated section to make the review article complete.

2-     2.2 paragraph, about the definition of CSCs, is not widely discussed. In my opinion, since the topic of the review article, it should be improved. I suggest revising 2.1 and 2.2 paragraphs thus to obtain a complete and exhausting definition and description of CSCs.

3-     In paragraph 2.3 (lines 102-105) the authors said that “CSCs from primary tumors or cancer cell lines are identified based on the expression of fluorescence or magnetic-beads-conjugated CSCs-markers. Then, the CSCs are sorted and examined for in vitro characteristics and also the formation of tumors in mouse models”, without mentioning these markers even not in the caption of Figure 2. For this reason, I suggest discussing Figure 2 more in paragraph 2.4 than in the relative caption.

4-     Often along the text, references are reported both with numbers and with authors’ names. Please modify them to ensure that the reference style is homogeneous, according to “Biomolecules” guidelines.

5-     Some paragraphs should be improved: even though many references are cited, the authors seem to report indications about where to find useful information without comments and report only the source from which to find them.

6-     About molecular markers, the authors reported only CD44, ALDH, and CD133 without deepening enough in OSCC/HNSCC. The review aims to report on OSCC, but it seems to be based on several other cancers too. The authors should strengthen this aspect focusing mainly on OSCC as expected.

7-     In paragraph 2.4.4, the authors reported about the epithelial-mesenchymal transition (EMT), but it is not clear its involvement in OSCC CSCs because once again the paragraph appears confused and not focused on OSCC. Moreover, the next paragraph about the signaling mechanisms of CSCs is interlaced with the previous one. The authors can consider revising and combining them, or dividing paragraph 3 according to the pathways considered.

8-     The authors reported a paragraph entitled “Cancer stem cells in therapy” only from an immunological point of view proposing several immunotherapy strategies. Did they find any other mechanisms or strategies, based on drugs, chemotherapy, or others, to avoid CSCs growth?

9-     Tables 1 and 2 reported relative references independently, making it difficult to address the papers between the text and the tables. My suggestion is to create a unique references list as reported at the end of the manuscript including both text and tables citations. In this light, “No”, “Authors” and “Citation/ID” can be removed from tables and only the number of the total reference list can be reported. About Table 2 the fifth column has not a title. Anyway, tables seem not to be connected with the text.

10-  Paragraph 5 includes a discussion about 3D-cell-based structures, like spheroids. Since the authors dedicated a specific paragraph of the review article to “In vivo evidence of CSCs” and “In vitro characteristics of CSCs”, why not add one to OSCC 3D-cell-based structures? Moreover, since the title is about OSCC, I do not agree with the authors to report about other cancers, such as human hepatoma, AML, or sarcoma.

Author Response

Dear authors, I carefully read and revised the manuscript entitled “Strategies for targeting Cancer Stem Cells in Oral Squamous Cell Carcinoma: A Comprehensive Review”. The review article aims to resume the current knowledge about cancer stem cells in Oral Squamous Cell Carcinoma, and the possible strategies to avoid recurrence and metastasis. In my opinion, the manuscript is not focused on the topic proposed by the title: paragraphs are not always clearly discussed and sometimes the purpose of the work is lost. For these reasons, I reported some suggestions to improve the review.

Responses: We sincerely thank the reviewer for the valuable feedback. We acknowledge the concern regarding the manuscript’s alignment with the proposed title. After careful consideration, we have revised the title to better reflect the focus and content of the review. The new title is “ Cancer Stem Cells in Oral Squamous Cell Carcinoma: A Narrative Review on Experimental Characteristics and Methodological Challenges.” This revision aims to clarify the scope and central theme of the manuscript, ensuring better coherence with the content discussed throughout the review.

  1. The review article lacks a methods paragraph in which it is possible to determine the search strategy used, and the number of studies evaluated or selected. I suggest the authors add a dedicated section to make the review article complete.

Responses: We appreciate the reviewer’s suggestion regarding the inclusion of a methods section. We agree that a clear description of the search strategy is essential for the completeness and transparency of the review. In response, we have added a dedicated paragraph detailing the search strategy used, including the databases consulted, keywords applied, and the criteria for study selection and evaluation. This information is now included in paragraph 1.1, lines 75-86.

  1. 2.2 paragraph, about the definition of CSCs, is not widely discussed. In my opinion, since the topic of the review article, it should be improved. I suggest revising 2.1 and 2.2 paragraphs thus to obtain a complete and exhausting definition and description of CSCs.

Responses: We acknowledge the reviewer’s valuable critique regarding the limited discussion in paragraph 2.2 on the definition of CSCs. Given that this is a key focus of the review, we have revised and expanded paragraphs 2.1 and 2.2 to provide a more comprehensive and thorough definition and description of CSCs. These revisions can be found at lines 88-109 and 110-123, respectively.

  1. In paragraph 2.3 (lines 102-105) the authors said that “CSCs from primary tumors or cancer cell lines are identified based on the expression of fluorescence or magnetic-beads-conjugated CSCs-markers. Then, the CSCs are sorted and examined for in vitro characteristics and also the formation of tumors in mouse models”, without mentioning these markers even not in the caption of Figure 2. For this reason, I suggest discussing Figure 2 more in paragraph 2.4 than in the relative caption.

Responses: We appreciate the reviewer’s insightful comment regarding the need to elaborate on the markers used to identify CSCs, particularly in relation to paragraph 2.3 and Figure 2. We have addressed this by expanding the discussion in paragraph 2.4 to include a detailed explanation of the specific markers involved, rather than only mentioning them in the figure caption. These revisions aim to enhance clarity and provide a more thorough understanding for readers. These revisions can be found at section 2.3 lines 135-137.

  1. Often along the text, references are reported both with numbers and with authors’ names. Please modify them to ensure that the reference style is homogeneous, according to “Biomolecules” guidelines.

Responses: We thank the reviewer for bringing this to our attention. We have carefully reviewed the manuscript and ensured that all references are now consistently formatted according to the "Biomedicines" guidelines. Specifically, we have standardized the citation style to use only numbered references throughout the text, eliminating any inconsistencies with author names.

  1. Some paragraphs should be improved: even though many references are cited, the authors seem to report indications about where to find useful information without comments and report only the source from which to find them.

Responses: We appreciate the reviewer’s observation regarding certain paragraphs that primarily list references without providing sufficient commentary or analysis. We have carefully revisited these sections and revised them to include more detailed discussions and critical evaluations of the cited sources. Our aim was to ensure that the text provides meaningful insights and not just references to where information can be found. These improvements enhance the depth and relevance of the content. Some evidence of these improvements can be found on section 2.4, line 160-225, line 231-280, line 388-523, line 599-706, where we have expanded our analysis and provided clearer commentary. 

  1. About molecular markers, the authors reported only CD44, ALDH, and CD133 without deepening enough in OSCC/HNSCC. The review aims to report on OSCC, but it seems to be based on several other cancers too. The authors should strengthen this aspect focusing mainly on OSCC as expected.

Responses: We thank the reviewer for the valuable feedback. We have addressed your concerns by focusing primarily on OSCC in our revised manuscript. Specifically, we have enhanced the discussion on CD44 (Section 2.4.3a at lines 203-231), ALDH (Section 2.4.3b at lines 232-257), and CD133 (Section 2.4.3c at lines 259-280) to better align with the focus on OSCC.

  1. In paragraph 2.4.4, the authors reported about the epithelial-mesenchymal transition (EMT), but it is not clear its involvement in OSCC CSCs because once again the paragraph appears confused and not focused on OSCC. Moreover, the next paragraph about the signaling mechanisms of CSCs is interlaced with the previous one. The authors can consider revising and combining them, or dividing paragraph 3 according to the pathways considered.

Responses: We apologize for the confusion regarding the discussion of the epithelial-mesenchymal transition (EMT) and its relevance to OSCC CSCs. To improve clarity and focus, we have moved the EMT section to be part of section 3, “Signaling Mechanisms of CSCs.” Additionally, we have reorganized the content by dividing the signaling pathways into distinct subsections: section 3.1 “JAK/STAT Pathway,” section 3.2 “Wnt/β-catenin Pathway,” section 3.3 “PI3K/Akt/mTOR Pathway,” and section 3.4 “EMT.” These changes, reflected from lines 309-412, aim to enhance the coherence and flow of the discussion.

  1. The authors reported a paragraph entitled “Cancer stem cells in therapy” only from an immunological point of view proposing several immunotherapy strategies. Did they find any other mechanisms or strategies, based on drugs, chemotherapy, or others, to avoid CSCs growth?

Responses: We acknowledge the reviewer’s valuable comment. In response, we have expanded the discussion in the section “Cancer Stem Cells in Therapy” to include additional strategies beyond immunotherapy. Specifically, we have incorporated mechanisms and strategies based on drugs, such as potential inhibitors, and chemotherapy approaches involving agents like 5-FU and cisplatin that are used to target CSC growth. We have also summarized the findings from Table 2 within the text for better integration. These amendments can be found in section 4, from lines 414-522.

  1. Tables 1 and 2 reported relative references independently, making it difficult to address the papers between the text and the tables. My suggestion is to create a unique references list as reported at the end of the manuscript including both text and tables citations. In this light, “No”, “Authors” and “Citation/ID” can be removed from tables and only the number of the total reference list can be reported. About Table 2 the fifth column has not a title. Anyway, tables seem not to be connected with the text.

Responses: We thank the reviewer for the helpful suggestions. In response, we have made the following revisions:

  • Unified Reference List: We have removed the columns labeled “No,” “Authors,” and “Citation/ID” from Tables 1 (line 725) and 2 (line 728). Instead, we have replaced these with reference numbers from a unified reference list included at the end of the manuscript.
  • Table 2 Column Title: We have added a title to the fifth column in Table 2 to ensure clarity.
  • Connection with Text: To improve the integration of the tables with the manuscript, we have incorporated the findings from Table 1 into Section 2.4.3 (lines 196-280) and the findings from Table 2 into Section 4 (lines 413-522).

We hope these changes address your concerns and enhance the clarity and coherence of the manuscript.

  1. Paragraph 5 includes a discussion about 3D-cell-based structures, like spheroids. Since the authors dedicated a specific paragraph of the review article to “In vivo evidence of CSCs” and “In vitro characteristics of CSCs”, why not add one to OSCC 3D-cell-based structures? Moreover, since the title is about OSCC, I do not agree with the authors to report about other cancers, such as human hepatoma, AML, or sarcoma.

Responses: We thank the reviewer for the helpful suggestions. In response, we have added and further discussed the development and use of 3D cell culture system in OSCC in the understanding of CSCs-OSCC at section 5.1, line 621-706.

Reviewer 2 Report

Comments and Suggestions for Authors

The authors cite and discuss many papers on CSCs in oral cancer in this manuscript, which has a certain academic value.

on the other hand, many review article described about CSCs in oral cancer (including head and neck cancer) already have existed, the novelty of this article is kind of questionable.

I believe that this review may be worthy of acceptance if significant revisions are made to the following points.

1)     The authors entitled this article as “Strategies for Targeting Cancer Stem Cells in Oral Squamous Cell Carcinoma: A Comprehensive Review”, but main contents of their manuscript are just description about experimental (in vivo/ vitro) characteristics of CSCs. Their title seems somewhat hyperbolic.

If they title it "Strategies for Targeting", they should talk and cite more articles about the origin of CSCs (the stem cells from which cancer stem cells are derived, and what kind of stem cells they are) and therapeutic strategies for CSCs.

Otherwise, they should change the title to “Experimental Characteristics of Cancer Stem Cells in Oral Squamous Cell Carcinoma”.

2)     In The section "2.1 The CSCs model" and Figure 1, they show two models of carcinogenesis, but for the CSCs model, "Jordan et al. NEJM 355; 1253, 2006" and cite it additionally and add a description. Then they should mention Nowell's name both within the Figure 1 figure and in the Figure legend.

Comments on the Quality of English Language

The descriptions within each section are kind of bulleted. I understand the impact of this being a “review”, but it would be easier for the author to read if they improved the connections between sentences a bit more, or if they focused on what they want to say in a section and stated the sentences more smoothly.

Author Response

The authors cite and discuss many papers on CSCs in oral cancer in this manuscript, which has a certain academic value.

On the other hand, many review article described about CSCs in oral cancer (including head and neck cancer) already have existed, the novelty of this article is kind of questionable.

I believe that this review may be worthy of acceptance if significant revisions are made to the following points.

  1. The authors entitled this article as “Strategies for Targeting Cancer Stem Cells in Oral Squamous Cell Carcinoma: A Comprehensive Review”, but main contents of their manuscript are just description about experimental (in vivo/ vitro) characteristics of CSCs. Their title seems somewhat hyperbolic. If they title it "Strategies for Targeting", they should talk and cite more articles about the origin of CSCs (the stem cells from which cancer stem cells are derived, and what kind of stem cells they are) and therapeutic strategies for CSCs. Otherwise, they should change the title to “Experimental Characteristics of Cancer Stem Cells in Oral Squamous Cell Carcinoma”.

Response: We agree with the reviewer’s comment regarding the discrepancy between the manuscript’s content and the original title. In line with this feedback, we have revised the title to better reflect the focus of the manuscript. The new title is: “Cancer Stem Cells in Oral Squamous Cell Carcinoma: A Narrative Review on Experimental Characteristics and Methodological Challenges.” This change ensures the title accurately represents the scope and content of the review.

  1. In The section "2.1 The CSCs model" and Figure 1, they show two models of carcinogenesis, but for the CSCs model, "Jordan et al. NEJM 355; 1253, 2006" and cite it additionally and add a description. Then they should mention Nowell's name both within the Figure 1 figure and in the Figure legend.

Response: We thank the reviewer for the suggestion. The amendment can be found on section 2, line 107-110.

  1. Comments on the Quality of English Language

The descriptions within each section are kind of bulleted. I understand the impact of this being a “review”, but it would be easier for the author to read if they improved the connections between sentences a bit more, or if they focused on what they want to say in a section and stated the sentences more smoothly.

Response: We appreciate the reviewer’s feedback regarding the flow and readability of the manuscript. We understand that some sections may come across as disjointed or overly segmented. To improve the overall coherence, we have carefully revised the text to enhance the connections between sentences and ensure smoother transitions within each section. Our revisions focus on maintaining a clear narrative while preserving the informative nature of the review. These improvements aim to make the manuscript more readable and engaging for the audience.

Reviewer 3 Report

Comments and Suggestions for Authors

This manuscript deals with the identification of cancer stem cells in oral squamous cell carcinomas (OSCC). Also, the therapeutic targeting of these cells has been considered.

The authors describe in detail the phenotypic and functional features of cancer stem cells (CSCs) in general and in particular in OSCC patients.

The review is well organized and presented. The flow of information and the content of this review is good and easy to understand.

The main matter is always how is it possible to consider some markers as typical markers of CSCs (and in this case, CSCs of OSCC) when these molecules can be present on several kinds of cells (CD44 and CD133 are expressed at different level in different tumor cell populations). This point should be explained well to allow the readers to understand that the markers used do not fit well with CSCs  or at least they do not fit only with CSCs.

There are some references that should be considered and discussed in this manuscript:

Aquino IG, Cuadra-Zelaya FJM, Bizeli ALV, Palma PVB, Mariano FV, Salo T, Coletta RD, Bastos DC, Graner E. Isolation and phenotypic characterization of cancer stem cells from metastatic oral cancer cells. Oral Dis. 2024 May 20. doi: 10.1111/odi.15003. Epub ahead of print. PMID: 38764396.

Huang J, Li H, Yang Z, Liu R, Li Y, Hu Y, Zhao S, Gao X, Yang X, Wei J. SALL4 promotes cancer stem-like cell phenotype and radioresistance in oral squamous cell carcinomas via methyltransferase-like 3-mediated m6A modification. Cell Death Dis. 2024 Feb 14;15(2):139. doi: 10.1038/s41419-024-06533-9. PMID: 38355684; PMCID: PMC10866932.

Liu C, Zhou S, Tang W. USP14 promotes the cancer stem-like cell properties of OSCC via promoting SOX2 deubiquitination. Oral Dis. 2024 Feb 20. doi: 10.1111/odi.14896. Epub ahead of print. PMID: 38376172.

Ortiz RC, Amôr NG, Saito LM, Santesso MR, Lopes NM, Buzo RF, Fonseca AC, Amaral-Silva GK, Moyses RA, Rodini CO. CSChighE-cadherinlow immunohistochemistry panel predicts poor prognosis in oral squamous cell carcinoma. Sci Rep. 2024 May 8;14(1):10583. doi: 10.1038/s41598-024-55594-5. PMID: 38719848; PMCID: PMC11078993.

de Castro LR, de Oliveira LD, Milan TM, Eskenazi APE, Bighetti-Trevisan RL, de Almeida OGG, Amorim MLM, Squarize CH, Castilho RM, de Almeida LO. Up-regulation of TNF-alpha/NFkB/SIRT1 axis drives aggressiveness and cancer stem cells accumulation in chemoresistant oral squamous cell carcinoma. J Cell Physiol. 2024 Feb;239(2):e31164. doi: 10.1002/jcp.31164. Epub 2023 Dec 27. PMID: 38149816.

Bai J, Chen Y, Sun Y, Wang X, Wang Y, Guo S, Shang Z, Shao Z. EphA2 promotes the transcription of KLF4 to facilitate stemness in oral squamous cell carcinoma. Cell Mol Life Sci. 2024 Jun 25;81(1):278. doi: 10.1007/s00018-024-05325-w. PMID: 38916835.

Todoroki K, Abe Y, Matsuo K, Nomura H, Kawahara A, Nakamura Y, Nakamura M, Seki N, Kusukawa J. Prognostic effect of programmed cell death ligand 1/programmed cell death 1 expression in cancer stem cells of human oral squamous cell carcinoma. Oncol Lett. 2024 Jan 4;27(2):79. doi: 10.3892/ol.2024.14213. PMID: 38249811; PMCID: PMC10797318.

Comments on the Quality of English Language

English language is good

Author Response

This manuscript deals with the identification of cancer stem cells in oral squamous cell carcinomas (OSCC). Also, the therapeutic targeting of these cells has been considered.

The authors describe in detail the phenotypic and functional features of cancer stem cells (CSCs) in general and in particular in OSCC patients.

The review is well organized and presented. The flow of information and the content of this review is good and easy to understand.

  1. The main matter is always how is it possible to consider some markers as typical markers of CSCs (and in this case, CSCs of OSCC) when these molecules can be present on several kinds of cells (CD44 and CD133 are expressed at different level in different tumor cell populations). This point should be explained well to allow the readers to understand that the markers used do not fit well with CSCs or at least they do not fit only with CSCs.

Response: We appreciate the reviewer’s feedback. This is indeed a focus of this narrative review. The same doubt has been raised by others regarding the use of markers to identify CSCs [reference #24] in the manuscript. To the best of our knowledge and review, only CD44 was demonstrated to play a role in promoting CSCs traits and not just merely as a marker. This is discussed in section 2.4.3, line 196-280

  1. There are some references that should be considered and discussed in this manuscript:

Aquino IG, Cuadra-Zelaya FJM, Bizeli ALV, Palma PVB, Mariano FV, Salo T, Coletta RD, Bastos DC, Graner E. Isolation and phenotypic characterization of cancer stem cells from metastatic oral cancer cells. Oral Dis. 2024 May 20. doi: 10.1111/odi.15003. Epub ahead of print. PMID: 38764396.

Huang J, Li H, Yang Z, Liu R, Li Y, Hu Y, Zhao S, Gao X, Yang X, Wei J. SALL4 promotes cancer stem-like cell phenotype and radioresistance in oral squamous cell carcinomas via methyltransferase-like 3-mediated m6A modification. Cell Death Dis. 2024 Feb 14;15(2):139. doi: 10.1038/s41419-024-06533-9. PMID: 38355684; PMCID: PMC10866932.

Liu C, Zhou S, Tang W. USP14 promotes the cancer stem-like cell properties of OSCC via promoting SOX2 deubiquitination. Oral Dis. 2024 Feb 20. doi: 10.1111/odi.14896. Epub ahead of print. PMID: 38376172.

Ortiz RC, Amôr NG, Saito LM, Santesso MR, Lopes NM, Buzo RF, Fonseca AC, Amaral-Silva GK, Moyses RA, Rodini CO. CSChighE-cadherinlow immunohistochemistry panel predicts poor prognosis in oral squamous cell carcinoma. Sci Rep. 2024 May 8;14(1):10583. doi: 10.1038/s41598-024-55594-5. PMID: 38719848; PMCID: PMC11078993.

de Castro LR, de Oliveira LD, Milan TM, Eskenazi APE, Bighetti-Trevisan RL, de Almeida OGG, Amorim MLM, Squarize CH, Castilho RM, de Almeida LO. Up-regulation of TNF-alpha/NFkB/SIRT1 axis drives aggressiveness and cancer stem cells accumulation in chemoresistant oral squamous cell carcinoma. J Cell Physiol. 2024 Feb;239(2):e31164. doi: 10.1002/jcp.31164. Epub 2023 Dec 27. PMID: 38149816.

Bai J, Chen Y, Sun Y, Wang X, Wang Y, Guo S, Shang Z, Shao Z. EphA2 promotes the transcription of KLF4 to facilitate stemness in oral squamous cell carcinoma. Cell Mol Life Sci. 2024 Jun 25;81(1):278. doi: 10.1007/s00018-024-05325-w. PMID: 38916835.

Todoroki K, Abe Y, Matsuo K, Nomura H, Kawahara A, Nakamura Y, Nakamura M, Seki N, Kusukawa J. Prognostic effect of programmed cell death ligand 1/programmed cell death 1 expression in cancer stem cells of human oral squamous cell carcinoma. Oncol Lett. 2024 Jan 4;27(2):79. doi: 10.3892/ol.2024.14213. PMID: 38249811; PMCID: PMC10797318.

Response: We thank the reviewer suggestion. All the references have been incorporated in this manuscript and the information can be found as below:

  1. Aquino IG, Cuadra-Zelaya FJM, Bizeli ALV, Palma PVB, Mariano FV, Salo T, Coletta RD, Bastos DC, Graner E. Isolation and phenotypic characterization of cancer stem cells from metastatic oral cancer cells. Oral Dis. 2024 May 20. doi: 10.1111/odi.15003. Epub ahead of print. PMID: 38764396. (can be found at line 498)
  2. Huang J, Li H, Yang Z, Liu R, Li Y, Hu Y, Zhao S, Gao X, Yang X, Wei J. SALL4 promotes cancer stem-like cell phenotype and radioresistance in oral squamous cell carcinomas via methyltransferase-like 3-mediated m6A modification. Cell Death Dis. 2024 Feb 14;15(2):139. doi: 10.1038/s41419-024-06533-9. PMID: 38355684; PMCID: PMC10866932. (can be found at line 149 and in table 1, ref 67)
  3. Liu C, Zhou S, Tang W. USP14 promotes the cancer stem-like cell properties of OSCC via promoting SOX2 deubiquitination. Oral Dis. 2024 Feb 20. doi: 10.1111/odi.14896. Epub ahead of print. PMID: 38376172. (can be found at line 358-361)
  4. Ortiz RC, Amôr NG, Saito LM, Santesso MR, Lopes NM, Buzo RF, Fonseca AC, Amaral-Silva GK, Moyses RA, Rodini CO. CSChighE-cadherinlow immunohistochemistry panel predicts poor prognosis in oral squamous cell carcinoma. Sci Rep. 2024 May 8;14(1):10583. doi: 10.1038/s41598-024-55594-5. PMID: 38719848; PMCID: PMC11078993. (can be found at line 254-257)
  5. de Castro LR, de Oliveira LD, Milan TM, Eskenazi APE, Bighetti-Trevisan RL, de Almeida OGG, Amorim MLM, Squarize CH, Castilho RM, de Almeida LO. Up-regulation of TNF-alpha/NFkB/SIRT1 axis drives aggressiveness and cancer stem cells accumulation in chemoresistant oral squamous cell carcinoma. J Cell Physiol. 2024 Feb;239(2):e31164. doi: 10.1002/jcp.31164. Epub 2023 Dec 27. PMID: 38149816. (can be found at line 498)
  6. Bai J, Chen Y, Sun Y, Wang X, Wang Y, Guo S, Shang Z, Shao Z. EphA2 promotes the transcription of KLF4 to facilitate stemness in oral squamous cell carcinoma. Cell Mol Life Sci. 2024 Jun 25;81(1):278. doi: 10.1007/s00018-024-05325-w. PMID: 38916835. (can be found at line 227-231)
  7. Todoroki K, Abe Y, Matsuo K, Nomura H, Kawahara A, Nakamura Y, Nakamura M, Seki N, Kusukawa J. Prognostic effect of programmed cell death ligand 1/programmed cell death 1 expression in cancer stem cells of human oral squamous cell carcinoma. Oncol Lett. 2024 Jan 4;27(2):79. doi: 10.3892/ol.2024.14213. PMID: 38249811; PMCID: PMC10797318. (can be found at line 229-231).

Round 2

Reviewer 1 Report

Comments and Suggestions for Authors

Dear authors, I read the revised version of the manuscript. In my opinion all the comments have been solved, even if the review appears long-winded. Anyway, I have no other comments to say.

Author Response

Dear Reviewer 1,

Thank you for taking the time to review our revised manuscript and for your thoughtful feedback. We are pleased to hear that you feel we have addressed all of the previous comments. We acknowledge your observation regarding the length of the review, and we will keep this in mind for future work. However, we aimed to provide a comprehensive and detailed discussion on the topic to thoroughly cover the complexities surrounding CSCs in OSCC.

We appreciate your valuable insights throughout the review process and are grateful for your contribution to improving the quality of our manuscript.

Reviewer 2 Report

Comments and Suggestions for Authors

Frankly speaking, I don't see much novelty in this article, but the authors have carefully summarized it. I think the illustrations of mice is too tedious, so I think it would be better to describe it briefly in words, not illustrations. If this is improved, it would be worthy of acceptance.

Comments on the Quality of English Language

N/A

Author Response

Reviewer 2

Frankly speaking, I don't see much novelty in this article, but the authors have carefully summarized it. I think the illustrations of mice is too tedious, so I think it would be better to describe it briefly in words, not illustrations. If this is improved, it would be worthy of acceptance.

Response:

We thank the reviewer for the constructive feedback. We agree that the illustration was redundant and have removed both the row for the illustration and the tumor growth data in Table 1. Additionally, we have updated the labeling of the site from "tongue" to "orthotopic tongue" to ensure readers understand that the inoculation was not subcutaneous.

We have made every effort to present novelty in our review, particularly in Sections 4 and 5, which we believe offer valuable insights. These sections address current methodological challenges and discuss future research directions to improve experimental systems and advance CSC studies. Section 4 also highlights evidence linking CSCs and their downstream signaling pathways to chemoresistance, and it introduces potential novel drug candidates aimed at overcoming CSC-induced chemoresistance in preclinical models of HNSCC.

Our review discusses current efforts to target CSC chemoresistance using agents such as hyaluronic acid (HA, anti-CD44), inhibitors of apoptosis proteins (IAPs), secreted frizzled-related protein 4 (sFRP4), curcumin, valproic acid (VPA), NCT-501, heat shock protein 90 (Hsp90) inhibitors, SB203580, XAV-939, SVC112, ketorolac, nuclear factor kappa-light-chain-enhancer of activated B cells (NF-κB) inhibitors, and TVB-3166. Through our literature review spanning the past 10 years, we also identified two novel drugs, casein kinase 2 (CK2) inhibitor and MEDI0641, which were found to suppress CSC subpopulations in HNSCC.

Furthermore, we emphasize the importance of understanding and targeting the crosstalk between immunosuppressive cells, cytokines, and CSCs to improve therapeutic outcomes. This is especially relevant as immune checkpoint inhibitors (ICIs) such as pembrolizumab and nivolumab have shown limited efficacy, which is partly due to immune evasion mechanisms mediated by CSCs. These CSCs contribute to immune suppression by impairing antigen presentation and promoting immunosuppressive cells, such as Tregs and MDSCs, that drive the PD-L1/PD-1 axis.

We have added this novelty in the abstract at line 36 and 37.